# Industry 4.0 Diagnosis from an iMillennial Educational Perspective

**Gabriela Beatrice Cotet [1,*]**, **Nicoleta Luminita Carutasu [2]** and **Florina Chiscop [2]**

[1] Training for Teaching Career and Social Sciences Department, University POLITEHNICA of Bucharest, 313, Splaiul Independenţei, Sector 6, RO-060042 Bucharest, Romania

[2] Robots and Production Systems Department, University POLITEHNICA of Bucharest, 313, Splaiul Independenţei, Sector 6, RO-060042 Bucharest, Romania; nicoleta.carutasu@upb.ro (N.L.C.); florina@mix.mmi.pub.ro (F.C.)

[*] Correspondence: gabriela.cotet@upb.ro

**Abstract:** Although the new economic paradigm is based on the rapid evolution of technology, it is not clear if this evolution is only dependent on a spectacular transformation of human resources or if the evolution of human resources has imposed major changes at a technical level as well. The main focus of this paper is to identify how to cope with these new technologies as educational actors, using a diagnosis of contemporary generation characteristics. The fourth industrial revolution (Industry 4.0) imposes a rapid evolution (or revolution) of the human resources paradigm in engineering: iMillennials should adapt to that paradigm, and the paradigm should be adapted to them. The research objectives were to identify some relevant characteristics of iMillennials' technological background and to create a map of the abilities of this generation as required by the evolution of new technologies. For a batch of students with a technical background, two psychological inventories that describe emotional intelligence and motivation acquisition were applied. Each inventory used focuses on certain features that describe motivational achievement (AMI) or emotional intelligence (EQ-I). Besides the motivational features, the AMI questionnaire also refers to socio-emotional abilities. A correlation between the parameters of the two inventories occurred. Three correlated parameters (assertiveness, reality testing, and commitment) were identified. Based on these results, a constellation map of soft skills was designed to match characteristic features of iMillennials with necessary competencies for an Industry 4.0 environment. Furthermore, this paper proposes a tool for educational actors to cope with these transformations based on the new technologies of Industry 4.0 and the characteristics of the iMillennials generation.

**Keywords:** iMillennials generation; soft skills; constellation map of capabilities for education; Industry 4.0

---

## 1. Introduction: Industry 4.0 and the iMillennials Generation

Over the last decade, the workplace market has changed considerably. Many of the jobs for which people have been educated and trained have changed significantly, and their configuration is determined by the emergence of new digital technologies [1–7]. In some industrial fields and beyond, artificial intelligence now performs specific tasks, forcing employees in these sectors to exercise different, unique and human skills [7–12]. Changes due to Industry 4.0 should not be ignored. The evolution of technology requires the attention of decision-makers from the political, economic and social levels, especially from the perspective of educational policies. The role of the human factor in future advanced manufacturing has great significance in the competition with artificial robotics and intelligence [2,10].

One of the significant changes in the fourth industrial revolution will be the introduction of cybernetic systems (CPSs) [3,4,12]. Virtual reality and speech recognition, as well as the use of augmented reality, will change how work is done [2]. Another example is the direct collaboration of robots and people working together without obstacles [7,12].

Almost four out of ten millennials (38%) declare that their organizations already achieve a large or fair amount of advanced automation, advanced connectivity, artificial intelligence or robotics to adequately fulfil the mechanical tasks or analyses previously done by humans. Meanwhile, nearly half (47%) say their employers use Industry 4.0 technologies to improve efficiency by increasing staff tasks or studies. Many millennials of Generation Z are already aware how Industry 4.0 is working in the labor market and prevision even more significant change [13].

We proposed the term "iMillennials" for young people in the new generation, replacing Generation Z, because we believe that the "i" joined to Millennials better characterizes this generation. The evolution of technology due to the development of the Internet and the speed at which this self-centered generation wants to connect to the real world have determined us to propose this term. This generation was born in 1995–2012. For us, iMillennials and Generation Z are the same, and the difference lies in the "i".

To focus of this research was to analyze the characteristics of iMillennials and identify the attributes corresponding to the requests of Industry 4.0 by assessing a batch of 120 students studying Industrial Engineering. Two inventories were used in this assessment, "Emotional Quotient Inventory"—EQ-I, and "Achievement Motivation Inventory"—AMI.

Dr. Reuven Bar-On's research on emotional intelligence has been conducted for over 20 years and has been tested by over 110,000 people worldwide. Bar-On's EQ-i® is the first inventory of emotional intelligence, which measures intelligent emotional and social behavior. One of the premises of the study is that emotional intelligence is an essential determinant of success in life. This inventory of emotional intelligence was formalized and applied to a group of students whose specialization is in the technical field. Emotional intelligence, as measured by this psychological inventory, refers to the emotional, personal, social and survival dimensions of intelligence, rather than one's ability to learn, think, reason or abstract. An emotional intelligence score helps to predict success in life. It also reflects current coping skills, one's ability to cope with daily environmental demands, common sense and, ultimately, general mental health. This psychological instrument consists of 133 items. For each statement, there are five possible answers: Very rarely or even never right for me—1, to Very often or even always right for me—5. The individual's responses render a total EQ score and scores on the following five composite scales. These comprise 15 subscale scores: Intrapersonal (involving Self-Regard, Emotional Self-Awareness, Assertiveness, Independence, and Self-Actualization); Interpersonal (comprising Empathy, Social Responsibility, and Interpersonal Relationship); Stress Management (containing Stress Tolerance and Impulse Control); Adaptability (providing Reality-Testing, Flexibility, and Problem-Solving); and General Mood (containing Optimism and Happiness). It is recommended for use in corporate settings for recruitment, screening, employee development and leadership programs [14].

The Achievement Motivation Inventory (AMI) is a personality inventory designed to measure a broad construct of work-related achievement motivation and the motivational traits that lead to socio-professional and also personal achievement. The motivation for accomplishment includes many facets that are closely linked to some traditional personality concepts. AMI is a psychological diagnostic tool that covers all the dimensions that are considered by one or more theorists as part of the motivation of achievement. In addition to the research forms in differential psychology and applied psychology, the major applications of AMI are for personnel selection, staff development, professional counseling, and so forth. It allows users to test candidates for 17 different facets of achievement motivation. For the first time, the essential social reasons from implementation are also integrated. AMI consists of 170 items to which the persons examined in a 7-point Likert format must respond: Not at all—1, to and Completely agree—7 [15]. A correlation between the parameters of the two inventories occurred, as explained later in this paper.

At first sight, the changes generated by the technological evolution create a very different generation from the one previously known. The real question is, however, if e-Revolution technology is shaping a new generation or if a new generation shapes the change at the technological level. The manufacturing industry continues to be a critical growth factor for economies around the world with a considerable contribution to trade, research and development and productivity, which accounts for 70% of exports and 90% of R&D investment in major manufacturing economies [16]. We can say that the skills and qualifications of the workforce are the keys to the success of an innovative factory, and the role of the human factor in future advanced manufacturing has great significance in the competition with artificial robotics and intelligence. To develop the potential of human resources in production, social actors represented by political power, education and industry have committed themselves to work together in efforts to create and develop their workforce.

Nowadays, production is in the middle of the fourth industrial revolution. Digitization inserts integrated and communication technologies into manufacturing architectures. With these new integrated systems, the future factory becomes adaptable in the production of small and unique batches tailored to the customer's requirements. Automation will become more and more critical, but a fully automatic factory is currently inadequate both in terms of control and economy. For the moment, it is only noticed that with the increase in the number of sensors used, there will be much more data extracted from the production processes, allowing for further analysis and optimization. These challenges will make support systems and services more critical in the future. The role of data analysis will be enhanced, thus helping factory employees to make smarter decisions and optimize processes.

One of the significant changes in the fourth industrial revolution will be the introduction of cybernetic systems (CPSs). CPSs are networks of interaction elements, including sensors, machine tools, assembly systems and parts, all connected through digital communication networks. The data collected by these networks will be virtually represented and the processes remote controlled. A CPS works as a system and is part of what is often called the Internet of Things (IoT), defined as a set of data and information stored in the virtual environment and accessible in integrated systems allowing the elements of these systems to communicate with each other [17–20]. According to Zuehlke, the Internet of Things "will probably be a non-deterministic and open network where self-organized intelligent entities (e.g., Web services) and virtual objects will be interoperable and capable of acting independently, following their objectives (or shared ones) context, circumstances or environment" [21].

The machines will communicate with each other, and the decentralized control systems will be able to optimize the production sequence. The manufacturing process will consist of small, standardized and combinable stages, where each product knows its way through the production sequence. There may be different products in the same production line, and machines and workers must be flexible in the event of changes in the production process. The work environment will move to the control or monitoring centers, where a qualified worker will have control over the manufacturing process. Virtual reality and speech recognition, as well as the use of augmented reality, will change how work is done. For example, displayed information or graphical instructions allow the user to see the work instructions during the critical activities [22–24].

The vision for the factory of the future begins to emerge. What is apparent is that the work done by qualified labor in a factory of the future will differ significantly from the situation in today's factories. The skills and competencies of the skilled workforce that are required to perform the tasks that are produced in a factory of the future will be different, modeled by interaction with robots and artificial intelligence [25–33]. This specification fits perfectly with the iMillennials typology in terms of the speed of information processing, primarily through visual techniques. Another example is the direct collaboration of robots and people working together without obstacles.

Figure 1 depicts the elements that the fourth industrial revolution intends to integrate into the development of new technologies: Internet of Things—IoT, Cyber-Physical Systems—CPSs, Artificial Intelligence—AI, Cloud Computing, Robotics and Additive Manufacturing. Industry 4.0 must rethink the role of robotics, 3D printing, artificial intelligence, nanotechnology and biotechnology. Also,

Industry 4.0 has pushed us toward a rethinking of wearable devices and augmented reality, the Internet of Things, Cloud Computing, and Autonomous Robots. The human factor is implicitly behind these new technologies, with their role not being directly linked to production but being especially linked to creation.

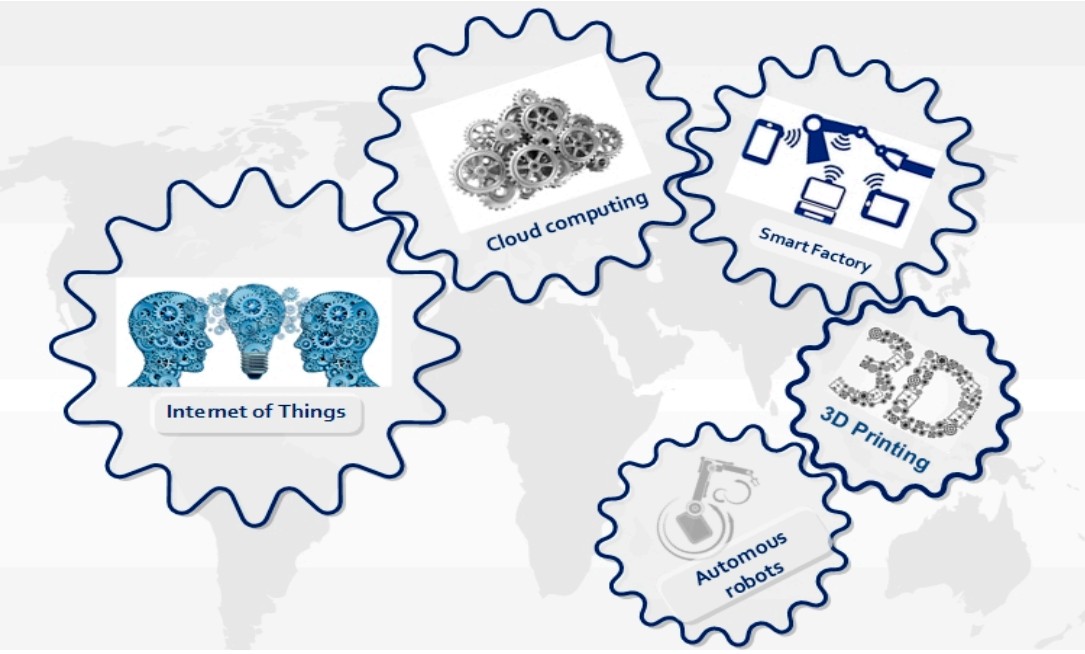

**Figure 1.** Industry 4.0: a vision of future industrial production.

On the one hand, according to Mannheim, the older generations form the social context with which the new generation comes into contact and is transformed. When this happens, the new generation changes social meaning by selecting or highlighting certain aspects. In conclusion, every new generation offers opportunities for continuity and social and cultural change for the next one [34].

It must be understood that new experiences are shaped by past experiences and are the source for shaping the future. The new spheres of learning (non-formal and informal learning, specifically lifelong learning) require a rethinking of learning strategies. According to the model developed by Strauss and Howe (1991), in our society, there is a generation change once every 20 years, with some signs of cyclicality [35].

It is essential to recognize the shades of human behavior in social groups to understand how small changes in attitude can lead to significant changes in the industrial paradigm.

For a generation that grows in full diversity and technological avalanche, innovations are no longer considered as disruptive. They represent the new normality. Virtual reality, self-driven cars and the ability to print almost everything using 3D technology do not surprise the members of Generation Z. They are only waiting for the next iteration of the process, and they want to play a part in designing it. It is the generation that went from expectation to innovation (Table 1).

**Table 1.** Specific characteristics for iMillennials [36–38].

| Characteristics/Meta-Factors | Particularity | Description |
|---|---|---|
| **Behaviors** | Goal orientation | This generation is motivated by social rewards, mentoring and constant feedback. They also want to make sense and be accountable. Like their predecessors, they also require flexible hours. |
| | Positive attitude | Dynamism, a bright career, inspiring work atmosphere and flexibility in work schedule. |
| | Technical experience | Generation Z is the most technologically gifted generation, connected at the global level. |
| | Multi-tasking | They can work efficiently with multiple tasks at the same time, with various sources of distraction in the background—multitasking. This kind of productivity flow could change how the activity is running. |
| | Global generation | Generation Z is the first truly global generation, primarily through technology, globalization and diverse cultural patterns, online entertainment, social trends, etc. |
| | Educational goal | The reformed reality from an educational point of view—moving from formal delivery to interactive environments. Therefore, education is no longer dependent on one stage of life but a lifelong reality. |
| **Attributes** | Low concentration | The iMillennials generation consumes many data. The result is a more rapid transmission of information, increasing the consumption of information, but with decreased concentration. |
| | Thinking outside the box | Ways of thinking differently, coming out of conventional patterns or matrices or from a new perspective. This expression often refers to new or creative, innovative thinking. It involves a process of thought focused on the implementation of an unusual approach to the structure of logical thinking. |
| | Taking career responsibility | iMillennials members seem to be more aware and concerned about their career. They are much more focused, determined and informed about career choice compared to Generation Y. |
| | Digital generation | They are digital integrators—the age at which we use technology for the first time determines how embedded it becomes in our lifestyle. |
| | Individualism | This is related to their competitiveness. They want to manage their projects so that their abilities shine. They do not want to depend on other people to do their job. |
| | Quick visual Interpretation | Transformation of visual interpretation pattern: image vs. word. Rapid processing of information. |
| **Values** | "Carpe diem" | They like to be involved in lawsuits, to contribute to finding solutions and being more engaged in various experiences. |
| | Meaningful work | The work carried out should be positive, deliberate and meaningful. Cultivating significance by focusing on the greater good of the work and by clearly showing the reason why the thing counts is a crucial practice researched to stimulate importance. |
| | Interpersonal relationships | The members of Generation Z revolt against personal interactions; human resources leaders should reassess how best to put the "human" aspect into business. For example, employment processes should focus on interviews in person more than online applications. |
| | Personal achievements | While all adult generations say that family is most important to their sense of self, the identity of Generation Z is best defined by their results. |

These are the main challenges that employers will likely face when they hire candidates from iMillennials: they want to change and provoke things, dogmas and best practices, and they want to explore and experiment [39,40].

## 2. Materials and Methods

The iMillennials generation believes that Industry 4.0 will increase the size of job offers, giving them more time to focus on creative, "human" and value-added work. One-fifth of them consider that

they will not be affected, while others think that the new paradigm of industry will be a severe threat. While only 17% of iMillennials expect Industry 4.0 to replace all jobs or some jobs, this increases to 32% for those whose organizations are extensively using Industry 4.0 technologies. These results suggest that familiarity with Industry 4.0 can inspire more fear than comfort [13].

Educational leaders should consider the features outlined above when they develop educational curricula to come to meet the expectations of this generation. Alongside educational leaders, industrial actors and political representatives involved in shaping social policies must participate.

Technical skills that will undoubtedly be useful, but not necessarily, are, for example, programming or coding abilities or not very profound technical knowledge. The future factory worker will be a generalist rather than a specialist [41].

Soft skills such as social and communication skills as well as teamwork and self-management skills become crucial. At present, the typical factory worker does not receive training in these areas because the job content usually does not require the use of these skills.

Our research can be regarded as a projection of what the future will be. The purpose of the study was, on the one hand, to diagnose the characteristic features of the iMillennials generation. On the other hand, our purpose was to map the competences that this generation has pursued, which will reshape the labor market, as we now know, and which will have to respond to the challenges posed by new technologies. The research was based on the application of two psychological instruments, the first based on emotional intelligence and the second based on the realization by motivation. The working hypothesis was that the values obtained for the representative scales would describe the characteristics considered relevant for this generation. Our interest has also focused on comparing the features described by other authors for iMillennials with the results obtained.

The values obtained by applicants for scales considered relevant for the characterization of iMillennials were considered in the two inventories: Achievement Motivation Inventory—AMI, and Emotional Quotient Inventory—EQ-I. The number of respondents was 120, students of the University Politehnica of Bucharest, with the specialization of industrial logistics and robotics.

According to the authors H. Schuler, GC Thorton III and Frintrup, AMI assesses the relevant factors for professional success. It goes from the premise that the motivation of realization results from the way in which a wide range of personality components are directed to performance. Its power is influenced by the following factors: the desire to establish and work towards achieving objectives—ambition; confidence in the ability to accomplish this—self-confidence; capacity to support efforts to attain set goals—self-control. Each of the above three factors is influenced by several aspects of one's personality [14]

The model for EQ-I is described as one of the most important approaches to emotional intelligence in Bar-On Emotional Quotient Inventory, TECHNICAL MANUAL (A Measure of Emotional Intelligence. This inventory of emotional intelligence was formalized and applied to a group of students whose specialization is in the technical field [15].

Following the analysis of the characteristics defining the iMillennials, we extracted a series of attributes that we have overlapped as the scales considered by us as relevant to the corresponding employee profile of the future that will be activated in Industry 4.0. Table 2 correlates the specific characteristics identified in the iMillennials generation (Table 1) with the scales we selected as relevant to the employee profile in Industry 4.0., from the two psychology inventories (EQ-I according to Bar-On, which describes emotional intelligence, and Achievement Motivation Inventory, which describes performance motivation).

For each critical attribute for the profile of the future employee, which characterizes iMillennials (left column), we have attached specific scales to the two psychological instruments AMI and EQ-I (middle column, respectively, right). We describe selected scales for AMI and EQ-I in Tables 3 and 4.

**Table 2.** The correlation between iMillennial competencies and selected scales from the Emotional Quotient Inventory (EQ-I) and the Achievement Motivation Inventory (AMI).

| Main Factors | AMI | EQ-I |
|---|---|---|
| Goal orientation | Status orientation | |
| Positive attitude | Flexibility<br>Confidence in success | Flexibility |
| Technical experience | Competitiveness | |
| Educational purpose | Status orientation | |
| Thinking outside of the box | Preference for difficult tasks<br>Fearlessness | Problem solving |
| Responsibility for careers | Engagement<br>Internality | Social responsibility |
| Individualism | Independence | Assertiveness<br>Independence |
| Carpe diem | Goal Setting | Reality testing |
| Meaningful work | | Optimism<br>Happiness |
| Personal achievement | Pride in productivity | Self-actualization |

**Table 3.** Description of the selected scales of the AMI inventory [15].

| | |
|---|---|
| Engagement | Identifies their availability to support an effort, its level and the individual workload. Actively engaged in their activity. Very active people, who give priority to work against other fields of business. |
| Confidence in success | Confidence in one's success describes the expectation of positive results from the activities one carries out. They act accordingly and expect to succeed, building on their skills and knowledge, even when faced with difficulties, obstacles or competition. |
| Flexibility | The way they relate to new situations and tasks. They have high availability for new professional conditions and prefer change or uncertainty. They are attracted to new locations that allow them to experience novelty, even if it means accepting a certain degree of discomfort or even the risk of failure. |
| Fearlessness | The people described by this scale are not afraid of failure. When faced with important tasks and situations, they do not feel a high level of pressure, and therefore their results are not negatively influenced. |
| Internality | Internality denotes how the results of one's actions are explained. They attribute the effects and consequences of their behavior to the actions taken. Depending on their response and effort, they interpret professional success or failure. |
| Preference for difficult tasks | They are described as ambitious, looking for challenges, ready for risk, self-questioning, wanting to prove their abilities, arduous, solving problems, stimulated by obstacles, testing their limits, and overcoming obstacles. Issues that arise are incentives rather than obstacles. |
| Independence | It is characterized as being responsible for oneself, autonomously, freely, and without obligations. They are involved in decision making; they are self-confident in the decision-making process. They want to decide on their way of working and make decisions independently. They do not like being controlled. |
| Status orientation | It describes the effort made to play an essential role in the social field and a leading place in the social hierarchy. Seeking recognition of their performance by others. They want to occupy influential positions and are interested in their professional promotion. Career growth prospects are, for them, essential motivators of professional performance. |
| Competitiveness | It measures the tendency to experience competition as an incentive and as a motivation for professional performance. People with high values are looking for match and comparison with others. They are quickly challenged; they want to be better and faster than others. They need to win. The gain strengthens them in their effort. |
| Goal setting | Setting goals for both short-term tasks and long-term projects. They are future-oriented and have high expectations of what they want to achieve. They have long-term plans, and they know in which direction they want to develop, from a personal point of view. |

**Table 4.** Description of the selected scales of the EQ-I inventory [14].

| | |
|---|---|
| AS: Assertiveness | Direct expression of one's own beliefs, thoughts or ideas. They use their own opinions for constructive purposes. Assertiveness involves finding the right language to support one's beliefs. |
| IN: Independence | Describes those who have high self-confidence. The importance of making the right decision in risk situations without being influenced by external factors. Identifies the persons who prefer being in a position to coordinate the activity in the team, taking initiative and responsibility. |
| SA: Self-actualization | The ability to realize their potential. Preference for learning new things that will bring them professional satisfaction. Identifies a long-term vision with motivational energy, and leads a meaningful life. |
| RE: Social responsibility | Ability to act responsibly. Socially responsible people accept, respect, and help others and respect social rules, also being able to use their skills to identify people with high moral and ethical standards. |
| RT: Reality testing | It characterizes people who can evaluate the relationship between what they experience (subjective) and what exists (objective). They are realistic people, looking for objective and pragmatic evidence to confirm, justify and support their feelings and actions. |
| FL: Flexibility | Rapid adaptation to unexpected situations, capable of responding to changes without rigidity. In search of new ideas and challenging, innovative situations in response to the problems they face. |
| PS: Problem-solving | A practical approach to solving the problems that arise, offering innovative solutions. The pleasure of facing risky situations. They have an individual ability to obtain intellectual and social resources. Heuristic and creative. |
| OP: Optimism | Positive attitude, which leads to high confidence in their strengths and themselves. They adapt quickly to troublesome or stressful situations. Optimistic people are generally desirable members of a team because they are more motivated and persevering. |
| HA: Happiness | The feeling of adequacy at work, in professional life, but also in leisure time and personal experience. They present a high degree of job satisfaction. They have a greater desire to explore, to seek new information and to have creative thinking. |

## 3. Results

The results obtained from the application of the two psychological instruments are shown in the graphs below.

The results obtained after applying the AMI, for the scales considered relevant for the study: Engagement—EN, Confidence in success—EZ, Flexibility—FX, Fearlessness—FU, Internality—IN, Preference for difficult tasks—SP, Independence—IN, Status orientation—ST, Competitiveness—WE and Goal setting—ZS, are presented in Figure 2. In the graph, the green color indicates the scores obtained for the selected scales, reported along with average scores for the same scales, presented in the chart with blue color.

The values are in the statistical range 90–110, described by the authors, H. Schuler, GC Thorton III and Frintrup as average motivational performance. The obtained values show us that there is an opportunity to develop an educational curriculum adapted to new paradigms imposed by Industry 4.0, to improve these traits.

The results obtained after applying AMI are in the average range of 90–110. This result shows us that there is a willingness to improve these traits.

The values obtained after applying the EQ-I inventory, for the scales relevant to the psychological profile required by the Industry 4.0 specification, are within the statistical range 90–109, described by Reuven Bar-On as efficient acting. This range represents the dynamic operating range (Figure 3).

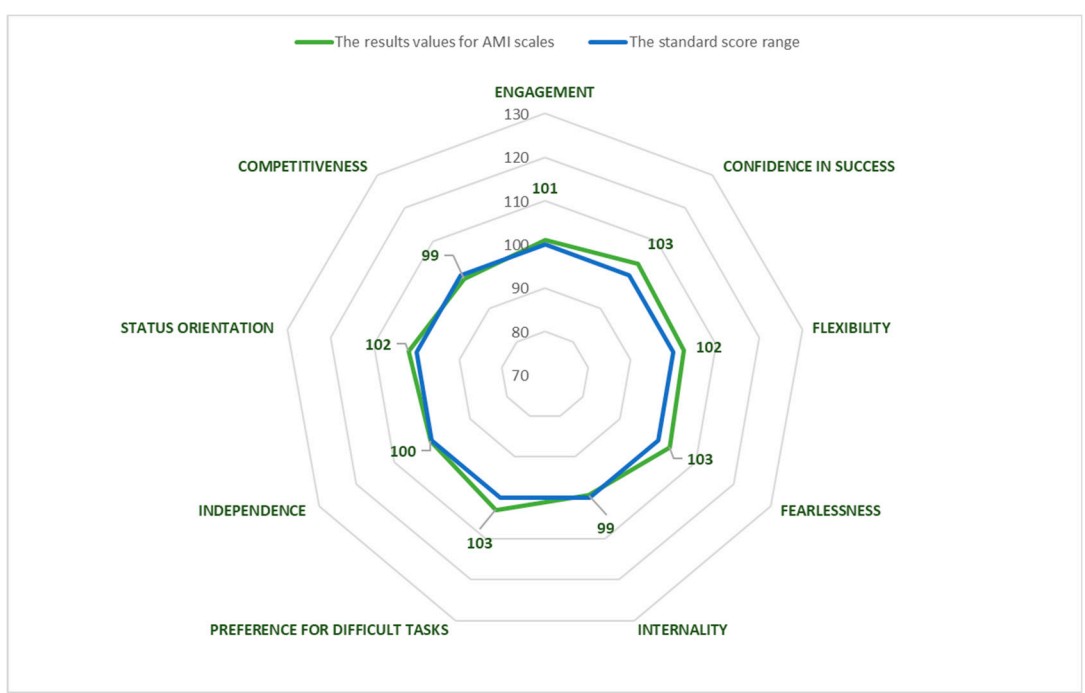

**Figure 2.** Selection of AMI scales correlated with the characteristics of iMillennials.

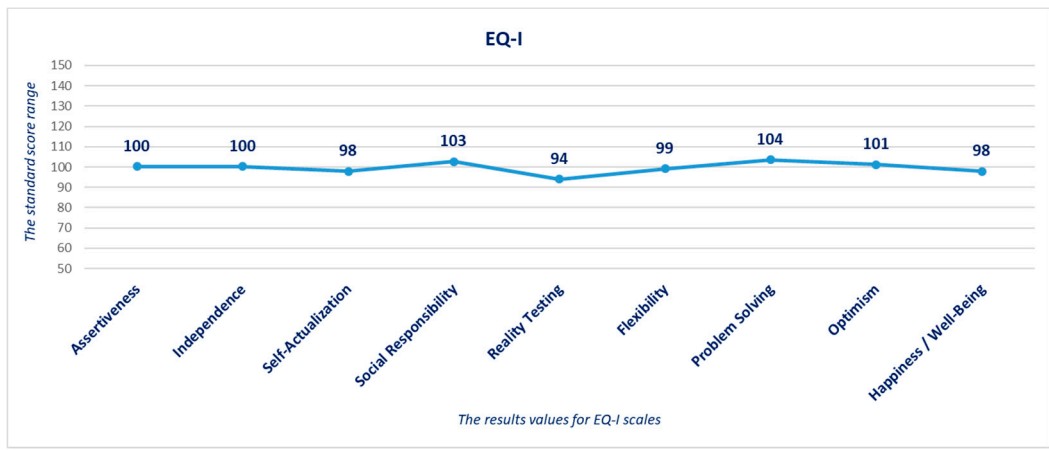

**Figure 3.** Selected EQ-I scales.

New technologies impose challenges with the rise of Industry 4.0. The intelligent systems associated with AI require from both of the actors involved in the development of new technologies, i.e., education and industry leaders, individual attention to ethics. There is a correlation between *engagement* and the *total coefficient of emotional intelligence—EQT* (general degree of adequate emotional and social functioning), the degree to which a person has specific noncognitive abilities and skills that they use successfully in adapting to the pressures and demands of the environment.

The values calculated by the Pearson coefficient indicate a correlation considered reasonable: 0.54 between engagement and assertiveness, 0. 49 between commitment and reality testing, and 0.41, respectively, for the total quotient of emotional intelligence (Figures 4 and 5).

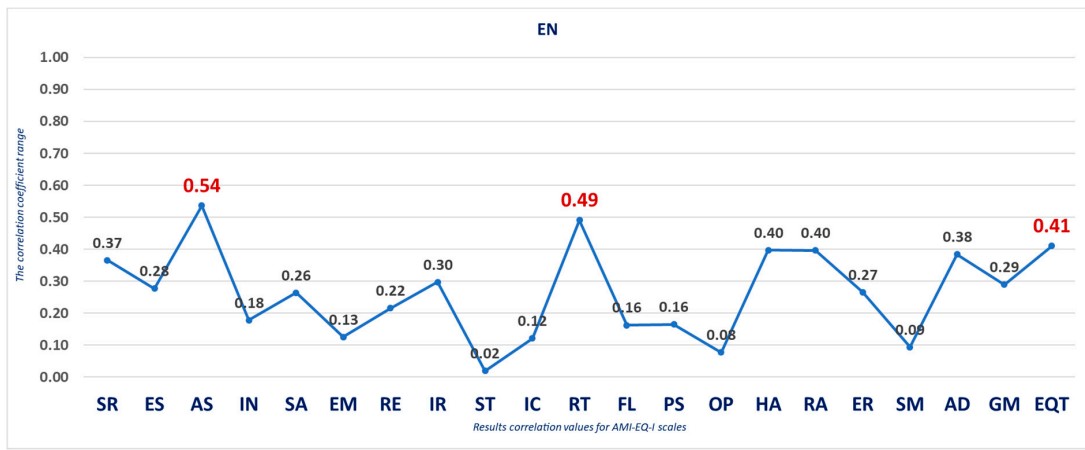

**Figure 4.** Graphical representation for the correlation between AMI and EQ-I.

|  | SR | ES | AS | IN | SA | EM | RE | IR | ST | IC | RT | FL | PS | OP | HA | RA | ER | SM | AD | GM | EQT |
|---|---|---|---|---|---|---|---|---|---|---|---|---|---|---|---|---|---|---|---|---|---|
| BE | 0.11 | 0.04 | 0.13 | -0.12 | 0.10 | -0.09 | 0.05 | 0.04 | 0.01 | -0.08 | 0.03 | -0.15 | 0.34 | 0.05 | 0.06 | 0.07 | 0.01 | -0.05 | 0.07 | 0.06 | 0.05 |
| DO | -0.14 | 0.20 | 0.14 | -0.30 | -0.20 | -0.03 | -0.02 | 0.08 | -0.24 | 0.01 | 0.08 | -0.12 | -0.28 | -0.33 | -0.07 | -0.12 | 0.02 | -0.11 | -0.11 | -0.24 | -0.14 |
| EN | 0.37 | 0.28 | 0.54 | 0.18 | 0.26 | 0.13 | 0.22 | 0.30 | 0.02 | 0.12 | 0.49 | 0.16 | 0.16 | 0.08 | 0.40 | 0.40 | 0.27 | 0.09 | 0.38 | 0.29 | 0.41 |
| EZ | -0.07 | 0.12 | 0.19 | -0.14 | -0.08 | 0.14 | 0.08 | 0.35 | -0.25 | -0.18 | 0.03 | -0.26 | 0.04 | -0.13 | 0.10 | -0.03 | 0.25 | -0.25 | -0.08 | -0.01 | -0.03 |
| FX | -0.03 | 0.03 | 0.19 | -0.23 | -0.07 | 0.04 | 0.10 | 0.13 | -0.23 | -0.03 | 0.20 | -0.04 | -0.03 | -0.15 | 0.17 | -0.05 | 0.11 | -0.14 | 0.08 | 0.02 | -0.01 |
| FL | 0.17 | 0.07 | 0.38 | 0.08 | 0.07 | -0.09 | 0.04 | 0.10 | -0.06 | 0.11 | 0.35 | 0.00 | -0.09 | -0.07 | 0.23 | 0.18 | 0.04 | 0.05 | 0.15 | 0.10 | 0.16 |
| FU | -0.21 | 0.03 | 0.00 | -0.29 | -0.30 | 0.06 | 0.15 | 0.14 | -0.13 | -0.13 | -0.12 | -0.27 | 0.12 | -0.18 | -0.17 | -0.22 | 0.14 | -0.16 | -0.14 | -0.21 | -0.17 |
| IN | -0.09 | -0.16 | -0.03 | -0.19 | 0.00 | 0.14 | 0.26 | 0.10 | -0.06 | -0.01 | -0.07 | -0.07 | -0.01 | 0.03 | 0.05 | -0.12 | 0.19 | -0.04 | -0.07 | 0.05 | -0.02 |
| KA | 0.21 | 0.02 | 0.32 | 0.07 | 0.23 | -0.10 | -0.04 | 0.07 | 0.07 | -0.04 | 0.12 | 0.17 | 0.10 | 0.15 | 0.27 | 0.21 | -0.01 | 0.01 | 0.17 | 0.26 | 0.18 |
| LS | 0.02 | -0.08 | 0.23 | -0.22 | 0.04 | 0.06 | 0.14 | 0.11 | -0.19 | -0.11 | 0.04 | -0.18 | -0.06 | -0.07 | 0.15 | -0.02 | 0.12 | -0.17 | -0.08 | 0.05 | -0.03 |
| LB | -0.02 | -0.11 | 0.31 | 0.03 | -0.10 | 0.28 | 0.36 | 0.20 | -0.11 | -0.06 | 0.18 | -0.18 | -0.09 | -0.03 | 0.06 | 0.00 | 0.32 | -0.10 | -0.02 | 0.02 | 0.06 |
| SP | 0.07 | 0.20 | 0.23 | -0.10 | 0.07 | 0.07 | 0.16 | 0.33 | -0.13 | -0.11 | 0.20 | -0.27 | 0.04 | -0.13 | 0.11 | 0.10 | 0.24 | -0.14 | 0.00 | 0.00 | 0.07 |
| SE | -0.08 | 0.17 | 0.19 | -0.27 | -0.25 | -0.08 | 0.02 | 0.31 | -0.27 | -0.11 | 0.09 | -0.24 | -0.24 | -0.34 | 0.03 | -0.10 | 0.13 | -0.22 | -0.14 | -0.18 | -0.13 |
| SK | -0.05 | -0.03 | 0.11 | -0.21 | -0.04 | 0.07 | 0.17 | 0.21 | -0.08 | 0.06 | 0.03 | 0.05 | 0.26 | -0.10 | 0.05 | -0.07 | 0.19 | 0.00 | 0.13 | -0.02 | 0.04 |
| ST | -0.09 | -0.16 | 0.05 | -0.23 | -0.03 | 0.06 | 0.14 | 0.11 | -0.26 | -0.23 | -0.12 | -0.34 | -0.16 | -0.16 | -0.01 | -0.12 | 0.13 | -0.29 | -0.27 | -0.10 | -0.17 |
| WE | 0.15 | -0.10 | 0.27 | 0.23 | 0.07 | -0.10 | -0.13 | -0.10 | 0.26 | -0.22 | -0.21 | 0.03 | 0.21 | 0.26 | -0.10 | 0.16 | -0.13 | -0.02 | -0.02 | 0.09 | 0.05 |
| ZS | -0.21 | 0.04 | 0.13 | -0.25 | -0.29 | 0.28 | 0.14 | 0.38 | -0.34 | -0.17 | -0.07 | -0.20 | -0.10 | -0.30 | 0.06 | -0.19 | 0.33 | -0.29 | -0.16 | -0.13 | -0.13 |
| IMG | 0.04 | 0.05 | 0.20 | -0.10 | 0.03 | -0.02 | 0.01 | 0.12 | -0.19 | -0.11 | 0.06 | -0.05 | 0.13 | -0.08 | 0.08 | 0.04 | 0.05 | -0.17 | 0.05 | 0.00 | 0.01 |

**Figure 5.** Results of the correlation between AMI and EQ-I scales.

According to the authors H. Schuler et al., Engagement—EN—describes the personal willingness to support an effort, its level and the workload. In other words, the ability to maintain a high level of activity over long periods. [15]

Following Reuven Bar-On, *assertiveness* represents the ability of a person to express their feelings, beliefs, and thoughts, and to defend their rights in a non-destructive manner. Assertiveness does not mean handling situations through high social and verbal skills but instead finding the proper language to support one's point of view [14].

Each inventory used focuses on specific features that describe motivational achievement (AMI) or emotional intelligence (EQ-I). Besides the motivational features, the AMI questionnaire also refers to socio-emotional abilities. A correlation between the parameters of the two inventories is described in Figures 4 and 5. We identified three correlated parameters (assertiveness, reality testing, and commitment). As one can see in Figure 5, these results fill the existing gap in terms of the correlation between engagement and total coefficient of emotional intelligence (the quotient given

by the general degree of emotional and social efficiency). Table 5 describes in detail the correlation between commitment, assertiveness, and reality testing.

**Table 5.** Correlations between AMI and EQ-I.

| AMI | EQ-I |
|---|---|
| **Commitment** | **Assertiveness** |
| An individual with a high score on the EN scale is described as an employee who is eager to work, diligent, busy, diligent, ambitious, enthusiastic for performance, dynamic, diligence, lively, action-oriented, entrepreneurial, agile, and restless, with ambitious skills. | Finding the right language to make others understand their point of view. Openly and constructively expresses their views. Expresses a categorical attitude to defend their beliefs, thoughts, and opinions, and express disapproval when they feel it. |
| | **Reality Testing** |
| | It is a characteristic of people able to assess the correspondence between what they experience (subjective) and what exists in reality (objective). |
| | Realistic persons, with their feet on the ground, trying to keep a refreshing perspective on the reality of things. Another feature of them is that they are always in search of objective and pragmatic evidence to confirm their ideas. |
| | They focus on examining ways they can cope with situations as they arise. Attention to the relevant information for the immediate solving of problems, taking better decisions and with fewer errors. |

## 4. Conclusions

In the technological context of Industry 4.0, the goal of an educational leader is to respond effectively to changing learning styles by implementing virtual and augmented reality, teaching as a facilitator, supporting group work, providing case studies in the real world and teaching through kinaesthetic learning methods. The skills mapping system proposed in this paper acts as an instrument to better understand what we want from iMillennials, as simple operators will be replaced by intelligent systems or creative employees, who must be speculators adapted to technological challenges.

Educational policies are the first source of modeling of the human resources of tomorrow. Competences imposed by new technologies will reshape the labor market as we know it. Today's students are continuously connected through advanced technologies and social networks. The education sector responds to this by adapting curricula and strategies to address the challenges and opportunities posed by this change.

The iMillennials generation is a globally connected generation in which people communicate in the virtual environment and industrial equipment communicates in the virtual environment as well. IoT technology opens the possibilities for creating sustainable economic growth, if educational policies will manage to ensure the competence of Generation Z to communicate with both people and industrial equipment with the same ease. In reducing the skills gap, it is essential to address the expectations coming from the new generations properly while also finding an appropriate response to the challenges of new technologies. Figure 6 describes the interaction of meta-factors and the main factors as given by the characteristics of iMillennials described in Table 1. The main elements are quantified by the scales of the two inventories, with the yellow color representing EQ-I, and the orange color AMI.

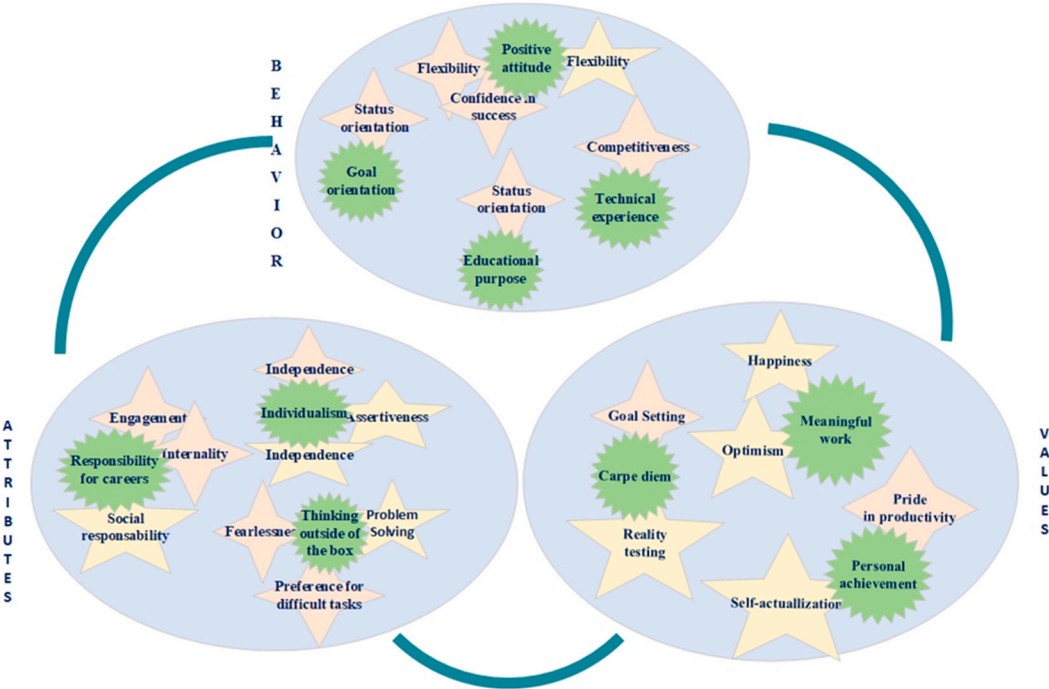

**Figure 6.** Constellation map of soft skills.

The focus was on identifying the level at which some main positive characteristics of iMillennials could be quantified to improve them through special programs. Lack of concentration is a negative element that was not quantified. For a group of students with a technical background, two psychological inventories that describe emotional intelligence and motivational acquisitions were applied. Each inventory used was focused on specific features that describe motivational achievement (AMI) or emotional intelligence (EQ-I). As a result, a correlation between three parameters (assertiveness, reality testing, and commitment) was determined. Based on these results, a constellation map of soft skills was produced matching characteristic features of iMillennials and necessary competencies for an Industry 4.0 environment. Starting from the vision of the main meta-factors, around which the secondary factors orbit, we defined the constellations of abilities and personal qualities—soft skills. We designed three meta-factors (primary valences): emotional intelligence, communication strategies, and thinking outside the box. Around each of these primary valences gravitates a series of substructures described as secondary valences. For each new job involved in the Industry 4.0 paradigm, certain specific secondary factors could be selected so as to fit the job description.

Further research will focus on the various applications of this constellation map of soft skills. We also intend to consider the inclusion of the negative features of iMillennials. The approach from this perspective will constitute one of the further developments of our research, especially from the standpoint of educational policies.

**Author Contributions:** Conceptualization, G.B.C. and N.L.C.; methodology, G.B.C.; software, F.C.; validation, G.B.C. and F.C.; formal analysis, G.B.C. and N.L.C.; investigation, G.B.C.; resources, F.C.; data curation, F.C.; writing—original draft preparation, G.B.C.; writing—review and editing, F.C. and N.L.C.; visualization, F.C. and N.L.C.; supervision, G.B.C.; project administration, G.B.C.; funding acquisition, F.C. All authors have read and agreed to the published version of the manuscript.

**Funding:** This work has been funded by University POLITEHNICA of Bucharest, through the "GNAC ARUT Program", UPB–GNAC ARUT. Identifier: UPB-GNAC ARUT-2018 Research project title: IOT platform material flow management in industry 4.0, Contract number: 162/ 01.10.2018.

**Acknowledgments:** This work has been funded by University POLITEHNICA of Bucharest, through the "GNAC ARUT Program". Identifier: UPB–GNAC ARUT Program–2018 Research project title: IOT platform material flow management in industry 4.0, Contract number: 162/ 01.10.2018.

**Conflicts of Interest:** The authors declare no conflict of interest. The funders had no role in the design of the study; in the collection, analyses, or interpretation of data; in the writing of the manuscript, or in the decision to publish the results.

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
