# Peer review of "Industry 4.0 Diagnosis from an iMillennial Educational Perspective"

_education, doi:10.3390/educsci10010021_

Round 1

Reviewer 1 Report

The paper "Industry 4.0 diagnosis from iMillennials education perspective" is an interesting paper. I find the topic modern, actual, and worth of scientific reseach. However after the reading of the paper I am satisfied only to some extent and I feel that certain corrections and improvements must be done if the article was to be published:

For someone unfamiliar with AMI and BARON EQ-I test (like me) it is unclear for me how the research was done and how the results were obtained. Please add some information about the construction of the above, and details about survey process. What does the obtained values in the range 90-110  (AMI) and 91 - 109 (EQ-I) exactly mean and how they were obtained? (In other word how the score is obtained, what is minimum/maximum level, are there certain score levels that should be interpreted in a certain way...) Please shed some more light on the construction and significance of Figure 4 and 5. The represented data shows correlations between results obtained by AMI & EQ-I surveys, however from the article I got the impression that above tools allow to measure different parameters. Please give information about labels on the Figure 4 and 5 as for now only some are explained in the text. Please explain how (why) parameters from AMI & EQ-I surveys were paired.  In my opinion it would be better to present description of AMI inventory (table 3), EQi inventory (Table 4) in the "Materials and Methods" instead of in the "Results" (please mind the 2nd comment about necesity of explaining all of AMI / EQi parameters shown on Figure 4 & 5). Althought English language and style are fine/minor spell check required Although I have selected "English language and style are fine/minor spell check required" please consider corrections of some parts of the article are written in a too-informal (for the scientific purposes) manner (for example "warp speed" in line 24). Also correct the interpunction errors (line 48 "now a days"; in table 1 add "." in few sentences; line 106 "friendly boss"; line 108 "." instead of ";"; Industry 4.0 line 111; develop in 117; for reference [26] and [27] put "." at the end; Line 207 "Assertiveness It does". Also it would be great if a native speaker checked the text. There is some concrete data presented in the article but without citation (lines 25-29, 39-41; 113-115). Please comment (briefly) Figure 1 Please correct the references accordingly to the template (usually "." is missing at the end).

Despite the above comments I would like to congratulate the author (authors?) creation of an interesting article. I am looking forward to see the improved version of it, and I am confident that author (authors?) should be fully capable to introduce suggested changes. If so the article in my opinion will be undoubtedly worthy the publication.

Author Response

Reviewer 1

For someone unfamiliar with AMI and BARON EQ-I test (like me) it is unclear for me how the research was done and how the results were obtained. Please add some information about the construction of the above, and details about survey process. What does the obtained values in the range 90-110 (AMI) and 91 - 109 (EQ-I) exactly mean and how they were obtained? (In other word how the score is obtained, what is minimum/maximum level, are there certain score levels that should be interpreted in a certain way...)

For a better understanding of the inventorys’ used (Bar-On EQ-I and AMI) we added some more detailed information in the paper:

“Emotional Quotient Inventory EQ-I: Dr. Reuven Bar-On's research on emotional intelligence has been conducted for over 20 years and has been tested by over 110,000 people worldwide. Bar-On EQ-i® is the first inventory of emotional intelligence, which measures intelligent emotional and social behavior. One of the premises of the study is that emotional intelligence is an essential determinant of success in life. This inventory of emotional intelligence was formalized and applied to a group of students whose specialization is in the technical field.  Emotional intelligence, as measured by this psychological inventory, refers to the emotional, personal, social and survival dimensions of intelligence, rather than one's ability to learn, think, reason or abstract. An emotional intelligence score helps to predict success in life. It also reflects current coping skills, one's ability to cope with daily environmental demands, common sense and, ultimately, general mental health. This psychological instrument consists of 133 items. For each statement there are five possible answers: Very rarely or even never true for me - 1 and Very often or even always true for me-5. The individual’s responses render a total EQ score and scores on the following 5 composite scales that comprise 15 subscale scores: Intrapersonal (comprising Self-Regard, Emotional Self-Awareness, Assertiveness, Independence, and Self-Actualization); Interpersonal (comprising Empathy, Social Responsibility, and Interpersonal Relationship); Stress Management (comprising Stress Tolerance and Impulse Control); Adaptability (comprising Reality-Testing, Flexibility, and Problem-Solving); and General Mood (comprising Optimism and Happiness). Is recommended for use in corporate settings for recruitment, screening, employee development and leadership programs.

Achievement Motivation Inventory – AMI

The Achievement Motivation Inventory (AMI) is a personality inventory designed to measure a broad construct of work-related achievement motivation, the motivational traits that lead to socio-professional, but also a personal achievement. The motivation for accomplishment includes many facets that are closely linked to some traditional personality concepts. AMI is a psychological diagnostic tool that covers all the dimensions that are considered by one or more theorists as part of the motivation of the achievement. In addition to the research applications in differential psychology and applied psychology, the major applications of AMI are for personnel selection, staff development, professional counselling, etc.

The Motivation Inventory for Achievement (AMI) is a personality inventory designed to measure a broad construct of work-related motivation. Allows users to test candidates for 17 different facets of achievement motivation. For the first time, the relevant social reasons from the point of view of implementation are also integrated. AMI consists of 170 items to which the persons examined in a 7-point Likert format must respond: Absolutely not at all -1 and Completely agree -7.”

Please shed some more light on the construction and significance of Figure 4 and 5. The represented data shows correlations between results obtained by AMI & EQ-I surveys, however from the article I got the impression that above tools allow to measure different parameters. Please give information about labels on the Figure 4 and 5 as for now only some are explained in the text. Please explain how (why) parameters from AMI & EQ-I surveys were paired. 

Each inventory used focuses on certain features that describe motivational achievement (AMI) or emotional intelligence (EQ-I). Besides the motivational features the AMI questionnaire also refers to socio-emotional abilities. As a result of our research we identified a correlation between the parameters of the two inventories as represented in figures 4 and 5. We identified 3 corelated parameters (assertiveness, testing reality and commitment).

In my opinion it would be better to present description of AMI inventory (table 3), EQi inventory (Table 4) in the "Materials and Methods" instead of in the "Results" (please mind the 2nd comment about necesity of explaining all of AMI / EQi parameters shown on Figure 4 & 5).

Thank you for your recommendation, we replaced tables 3 and 4 from chapter 5 Results into chapter 4 Materials and methods.

Although I have selected "English language and style are fine/minor spell check required" please consider corrections of some parts of the article are written in a too-informal (for the scientific purposes) manner (for example "warp speed" in line 24).

We changed the style into a more formal one.

The statement "warp speed" has been reformulated as follows:

“The evolution of technology requires the attention of the decision-makers from the political, economic and social level, especially from the perspective of the educational policies”.

Also correct the interpunction errors line 48 "now a days"; in table 1 add "." in few sentences; line 106 "friendly boss"; line 108 "." instead of ";"; Industry 4.0 line 111; develop in 117; for reference [26] and [27] put "." at the end; Line 207 "Assertiveness It does".

We made the suggested corrections and replaced them with red in text.

There is some concrete data presented in the article but without citation (lines 25-29, 39-41; 113-115).

The reference for lines 25-29 is the same as for line 30: [1]

https://www2.deloitte.com/content/dam/Deloitte/au/Documents/about-deloitte/hrc-millennial-survey-report-2018.pdf

The citation for lines 39-41 was added at bibliography: [2]

The citation for lines 113-115 was added at bibliography: [26]

Please comment (briefly) Figure 1

Figure 1 depicts the elements that the fourth industrial revolution intends to integrate into the development of new technologies: Internet of Things - IoT, Cyber-Physical Systems - CPS and, Artificial-Intelligence IA, Cloud Computing, Robotics and Additive Manufacturing. Industry 4.0 must rethink the role of robotics, 3D printing, artificial intelligence, nanotechnology and biotechnology. Also Industry 4.0 pushes to a rethinking of wearable devices and augmented reality, Internet of Thigs, Cloud Computing, Autonomous Robots. The human factor is implicitly behind these new technologies, their role not being directly linked to production but especially to creation.

Please correct the references accordingly to the template (usually "." is missing at the end).

We modified the references accordingly.

Reviewer 2 Report

Thank you very much for giving me an opportunity to review this manuscript. I did it with a great pleasure as the topic fully covers my research interest. In my opinion the issue covered by the paper analysed is important and topical, especially for the cognitive aspects.

Author /s  however, some additional efforts should made to improve the quality of the submission:

C1.  Abstract: is too short and has a lack of purpose of the study, the main methods or treatments applied,the article's main findings,indicate the main conclusions or interpretations

C2. Introduction: is not compatible with "Instructions for Authors" and Abstract.

Author/s should remember that:

1.The introduction should briefly place the study in a broad context and highlight why it is important. (In this paper not enough)

The introduction should define the purpose of the work and its significance, including specific hypotheses being tested (In this paper- lack of puprposes and hypotheses)

3.In the introduction the current state of the research field should be reviewed carefully and key publications cited (In this paper -in introduction -  only one reference!!!!!! lack of literature review!!! ).

The introduction should highlight controversial and diverging hypotheses when necessary. The introduction should briefly mention the main aim of the work and highlight the main conclusions (n this paper - none). Author should keep the introduction comprehensible to scientists working outside the topic of the paper.

C3.Section 1,2 and 3 should be aggregated in one block and extended to a more in-depth literature review about the:1- challeneges of the age of Industry 4.0. for society, 2-companies and 3-contemporary education system (especially3, In the introduction, there is no broader reference to education problems in the Industry 4.0 era).  

C4. It is worth explaining at the beginning in the introduction, briefly in parentheses or footnotes, key terms for the study: iMillenials , Z generation,AMI,EQI.

C5. The text should include an embedding for tables and drawings, e.g. (see Figure 1, line 84), (see: Table 1, line 103).

C6. In Figure 1, the vision of the future of production in erz Industry 4.0 lacks the human factor !!! and source of reference. People will still be there and their knowledge will determine the effective use of the elements indicated in the picture !!

C7. Standardize abbreviations: line 137 EQI, line 147- EQ-I, BARON EQ-I,BarOn EQ-I, line 150- Bar-On  Model,line 181,186,199,211,216 and so on: EQi

C8. Complete the references missing in the text: line 152: Spielberger's  Encyclopaedia  of  Applied  151

Psychology;

C9: p.4 Marerials and Methods: The research process methodology used is very poorly described. It is difficult to know how the research was carried out in real terms, how the occurrence of the examined features was assessed in the research sample. There is no detailed description of the research tools in p. 4. Materials and Methods.

10: In table 2: lack of sources; It is worth considering modifying the title of Table 2.Selected scales ... Are the scales visible in the table or something else? 11: On Figure 2 lacks one factor described further in the text "SP-Preference for difficult tasks". There are only 9 factors, there should be 10. (line 166-169) Very poor, simple and short description and commentary on results (Figure 2 , 3, 4,5,Table 5). It is worth supplementing. 12 Tables 3 and 4 should be included in section 4.Materials and Methods as a supplements to the description of the research tools, not in section 5. Results.

C13. Check the correctness of the assignment References 22 and 23. It seems that the author meant 26 and 27.

C.14. Conclusions very simple, underdeveloped. They do not contribute much to the development of the education system for iMillennials in the era of Industry 4.0. They do not respond to signals about the aims of work mentioned quite generally in Abstract and Introduction.

Developing them and enriching them with specific proposals for changes in the iMillennials generation system and education methods would significantly increase the value of work.

C 15. There is no legend for the description. Figure 6. It is not known whether graphics mean stronger meaning of some skills. Stars of different shapes can mislead the reader if they mean nothing. It is necessary to make up for this lack. It is also worth developing a comment to the map in Figure 6. A good comment with relevant recommendations on how to realistically use such ordered knowledge about the generation of iMillenials would increase the value and reception of work.

C16. In Conclusions, there was a reference to Educational Policies - line 226, previously, unfortunately, little was said about it. It is worth to develop this topic in a new, expanded version of the Introduction to tie it in a coherent way with the Conclusions.

C17. The literature review should be expanded and updated.

C18. Please check for some editorial errors, missing references. Moreover, inconsistency in text format and reference style. Please present the references according to Education Sciences guidelines.

I congratulate the author(s) for this not easy work of trying to  bridge several key areas and concepts from contemporary social life ,  and wish them the most success. Don't hesitate to create own ideas, strategies, tools, programs of education for iMillenials generation in the age of Industry 4.0

Best regards

Reviewer

Author Response

Reviewer 2 

C1.  Abstract: is too short and has a lack of purpose of the study, the main methods or treatments applied,the article's main findings,indicate the main conclusions or interpretations.

Some more details were inserted as follows:

“The objectives of this research were to identify some relevant characteristics of iMillennials technological background and to create a map of the abilities of this generation as required by the evolutions of the new technologies. For a batch of students with technical background, two psychological inventories that describe emotional intelligence and motivation acquisitions were applied. Each inventory used focuses on certain features that describe motivational achievement (AMI) or emotional intelligence (EQ-I). Besides the motivational features the AMI questionnaire also refers to socio-emotional abilities. A correlation between the parameters of the two inventories occurred. Three corelated parameters (assertiveness, testing reality and commitment) were identified. Based on these results the constellation map of soft skills was designed to match characteristic features of the iMillennials and necessary competences for an Industry 4.0 environment.”

C2. Introduction: is not compatible with "Instructions for Authors" and Abstract.

Author/s should remember that:

1.The introduction should briefly place the study in a broad context and highlight why it is important. (In this paper not enough)

The introduction should define the purpose of the work and its significance, including specific hypotheses being tested (In this paper- lack of puprposes and hypotheses)

3.In the introduction the current state of the research field should be reviewed carefully and key publications cited (In this paper -in introduction -  only one reference!!!!!! lack of literature review!!! ).

The introduction should highlight controversial and diverging hypotheses when necessary. The introduction should briefly mention the main aim of the work and highlight the main conclusions (n this paper - none). Author should keep the introduction comprehensible to scientists working outside the topic of the paper.

We improved the introduction and references. In order to enhance the scientific foundation of the paper the authors decided to include some more relevant paragraphs and references [1]-[12] (see below).

“Over the last decade workplaces' market has changed considerably. Many of the jobs for which people have been educated and trained have changed significantly, their configuration being determined by the emergence of new digital technologies [1]-[7]. In some industrial fields and beyond, artificial intelligence now performs specific tasks, forcing employees in these sectors to exercise different, unique and human skills [7]-[12].The role of the human factor in future advanced manufacturing has a great significance in the competition with artificial robotics and intelligence [2], [10].

One of the major changes in the fourth industrial revolution will be the introduction of cybernetic systems (CPS) [3], [4], [12]. Virtual reality and speech recognition, as well as the use of augmented reality, will change the way in which work is done [2]. Another example is the direct collaboration of robots and people working together without obstacles [7], [12]. Industry 4.0 must rethink the role of robotics, 3D printing, artificial intelligence, nanotechnology and biotechnology. Also, Industry 4.0 pushes to a rethinking of wearable devices and augmented reality, Internet of Thigs, Cloud Computing, Autonomous Robots. The human factor is implicitly behind these new technologies, their role not being directly linked to production but especially to creation.”

[1] Villani, V., Pini, F., Leali, F., Secchi, C., 2018. Survey on human–robot collaboration in industrial settings: Safety, intuitive interfaces and applications. Mechatronics. doi:10.1016/j.mechatronics.2018.02.009

[2] Michalos, G., Makris, S., Tsarouchi, P., Guasch, T., Kontovrakis, D. and Chryssolouris, G., 2015. Design considerations for safe human-robot collaborative workplaces. Procedia CIRP, 37, pp.248-253.

[3] Wang, L., Gao, R., Váncza, J., Krüger, J., Wang, X.V., Makris, S., Chryssolouris, G., 2019. Symbiotic human-robot collaborative assembly. CIRP Ann. doi:10.1016/j.cirp.2019.05.002

[4] Tsarouchi, P., Michalos, G., Makris, S., Athanasatos, T., Dimoulas, K. and Chryssolouris, G., 2017. On a human–robot workplace design and task allocation system. International Journal of Computer Integrated Manufacturing, 30(12), pp.1272-1279.

[5] Michalos, G., Makris, S., Spiliotopoulos, J., Misios, I., Tsarouchi, P. and Chryssolouris, G., 2014. ROBOPARTNER: Seamless human-robot cooperation for intelligent, flexible and safe operations in the assembly factories of the future. Procedia CIRP, 23, pp.71-76.

[6] Someshwar, R., Meyer, J. and Edan, Y., Models and methods for HR synchronization. IFAC Proceedings Volumes, 45(6), 2012, pp.829-834.

[7] Roy S, Edan Y, Investigating joint-action in short-cycle repetitive handover tasks: the role of giver versus receiver and its implications for human–robot collaborative system design. Int J Social Robot. 2018, pp 1-16 https://doi.org/10.1007/s12369-017-0424-9

[8] Tsarouchi, P., Matthaiakis, A.S., Makris, S. and Chryssolouris, G., 2017. On a human-robot collaboration in an assembly cell. International Journal of Computer Integrated Manufacturing, 30(6), pp.580-589.

[9] Makris, S., Tsarouchi, P., Matthaiakis, A.S., Athanasatos, A., Chatzigeorgiou, X., Stefos, M., Giavridis, K. and Aivaliotis, S., 2017. Dual arm robot in cooperation with humans for flexible assembly. CIRP Annals, 66(1), pp.13-16.

[10] Someshwar, R. and Edan, Y., Givers & receivers perceive handover tasks differently: Implications for human-robot collaborative system design. arXiv preprint arXiv:1708.06207, 2017

[11] Someshwar, R., Meyer, J. and Edan, Y., A timing control model for hr synchronization. IFAC Proceedings Volumes, 45(22), 2012. pp.698-703.

[12] Someshwar, R. and Kerner, Y., Optimization of waiting time in HR coordination. In IEEE International Conference on Systems, Man, and Cybernetics, 2013, pp. 1918-1923.

C3. Section 1,2 and 3 should be aggregated in one block and extended to a more in-depth literature review about the:1- challeneges of the age of Industry 4.0. for society, 2-companies and 3-contemporary education system (especially3, In the introduction, there is no broader reference to education problems in the Industry 4.0 era).  

Section 1,2 and 3 were aggregated in one block and extended to a more in-depth literature review (see above).

C4. It is worth explaining at the beginning in the introduction, briefly in parentheses or footnotes, key terms for the study: iMillenials , Z generation, AMI, EQI.

For clarify these terms, we inserted in the introduction the following explanations:

We proposed the term "iMillennials" for young people in new generation, replacing Z generation, because we believe that the "i" joined to Millennials better characterize this generation. The evolution of technology due to the development of the Internet, the speed this self-centred generation wants to connect to the real world, have determined us to propose this term. This generation was born between 1995-2012. For us, iMillennials and Z generation are the same, the difference in between is the “i”.

Achievement Motivation Inventory – AMI

The Achievement Motivation Inventory (AMI) is a personality inventory designed to measure a broad construct of work-related achievement motivation, the motivational traits that lead to socio-professional, but also a personal achievement. The motivation for accomplishment includes many facets that are closely linked to some traditional personality concepts. AMI is a psychological diagnostic tool that covers all the dimensions that are considered by one or more theorists as part of the motivation of the achievement. In addition to the research applications in differential psychology and applied psychology, the major applications of AMI are for personnel selection, staff development, professional counselling, etc.

The Motivation Inventory for Achievement (AMI) is a personality inventory designed to measure a broad construct of work-related motivation. Allows users to test candidates for 17 different facets of achievement motivation. For the first time, the relevant social reasons from the point of view of implementation are also integrated. AMI consists of 170 items to which the persons examined in a 7-point Likert format must respond: Absolutely not at all -1 and Completely agree -7.

Emotional Quotient Inventory EQ-I: Dr. Reuven Bar-On's research on emotional intelligence has been conducted for over 20 years and has been tested by over 110,000 people worldwide. BarOn EQ-i® is the first inventory of emotional intelligence, which measures intelligent emotional and social behavior. One of the premises of the study is that emotional intelligence is an essential determinant of success in life. This inventory of emotional intelligence was formalized and applied to a group of students whose specialization is in the technical field.  Emotional intelligence, as measured by this psychological inventory, refers to the emotional, personal, social and survival dimensions of intelligence, rather than one's ability to learn, think, reason or abstract. An emotional intelligence score helps to predict success in life. It also reflects current coping skills, one's ability to cope with daily environmental demands, common sense and, ultimately, general mental health. This psychological instrument consists of 133 items. For each statement there are five possible answers: Very rarely or even never true for me - 1 and Very often or even always true for me-5. The individual’s responses render a total EQ score and scores on the following 5 composite scales that comprise 15 subscale scores: Intrapersonal (comprising Self-Regard, Emotional Self-Awareness, Assertiveness, Independence, and Self-Actualization); Interpersonal (comprising Empathy, Social Responsibility, and Interpersonal Relationship); Stress Management (comprising Stress Tolerance and Impulse Control); Adaptability (comprising Reality-Testing, Flexibility, and Problem-Solving); and General Mood (comprising Optimism and Happiness). Is recommended for use in corporate settings for recruitment, screening, employee development and leadership programs.

The Achievement Motivation Inventory (AMI) is a personality inventory designed to measure a broad construct of work-related achievement motivation, the motivational traits that lead to socio-professional, but also a personal achievement. The motivation for accomplishment includes many facets that are closely linked to some traditional personality concepts. AMI is a psychological diagnostic tool that covers all the dimensions that are considered by one or more theorists as part of the motivation of the achievement. In addition to the research applications in differential psychology and applied psychology, the major applications of AMI are for personnel selection, staff development, professional counselling, etc.

C5. The text should include an embedding for tables and drawings, e.g. (see Figure 1, line 84), (see: Table 1, line 103).

The text was modified accordingly.

C6. In Figure 1, the vision of the future of production in erz Industry 4.0 lacks the human factor !!! and source of reference. People will still be there and their knowledge will determine the effective use of the elements indicated in the picture !!

Figure 1 depicts the elements that the fourth industrial revolution intends to integrate into the development of new technologies: Internet of Things - IoT, Cyber-Physical Systems - CPS and, Artificial-Intelligence IA, Cloud Computing, Robotics and Additive Manufacturing. The human factor is implicitly behind these new technologies, their role not being directly linked to production but especially to creation.

C7. Standardize abbreviations: line 137 EQI, line 147- EQ-I, BARON EQ-I,BarOn EQ-I, line 150- Bar-On  Model,line 181,186,199,211,216 and so on: EQi

 Abbreviations: EQ-I according to Reuven Bar-On theory were standardized.

C8. Complete the references missing in the text: line 152: Spielberger's  Encyclopaedia  of  Applied  151

“Spielberger's  Encyclopaedia  of  Applied” was replaced with Bar-On Emotional Quotient Inventory, TECHNICAL MANUAL (A Measure of Emotional Intelligence. It was a regrettable mistake.

C9: p.4 Marerials and Methods: The research process methodology used is very poorly described. It is difficult to know how the research was carried out in real terms, how the occurrence of the examined features was assessed in the research sample. There is no detailed description of the research tools in p. 4. Materials and Methods.

Following the analysis of the characteristics defining the iMillennials, we extracted a series of attributes that we have overlapped as the scales considered by us as relevant to the corresponding employee profile of the future that will activate in Industry 4.0. In table 2, we correlate the specific characteristics identified in the iMillennials generation with the scales we selected as relevant to the employee profile in Industry 4.0., from the two psychology inventories AMI and EQ-I.

We described the relevant scales for research, seeking to the specific abilities demanded by the technology of Industry 4.0 into table 3 and 4. We applied the inventories to 120 students to observe the value of parameters.

According to the construction of the two inventories, for AMI the results obtained are recorded in the range 90-100, according to the authors with medium motivational skills; for EQ-I the results obtained are entered in the statistical range 90 – 109, described the author as efficient acting

C10: In table 2: lack of sources; It is worth considering modifying the title of Table 2.Selected scales ... Are the scales visible in the table or something else?

Table 2 correlates the specific characteristics identified in the iMillennials generation (table 1) with the scales we selected as relevant to the employee profile in Industry 4.0., from the two psychology inventories (EQ-I according to Reuven Bar-On that describes emotional intelligence and Achievement Motivational Inventory that describes performance motivation)

C11: On Figure 2 lacks one factor described further in the text "SP-Preference for difficult tasks". There are only 9 factors, there should be 10. (line 166-169) Very poor, simple and short description and commentary on results (Figure 2 , 3, 4,5,Table 5). It is worth supplementing.

Thank you very much for your observation, we made the modification into the graph and added the lost scale “SP-Preference for difficult tasks”.

C12 Tables 3 and 4 should be included in section 4.Materials and Methods as a supplements to the description of the research tools, not in section 5. Results.

Thank you for your recommendation, we moved tables 3 and 4 from chapter 5 Results into chapter 4 Materials and methods.

C13. Check the correctness of the assignment References 22 and 23. It seems that the author meant 26 and 27.

The numbering of the bibliographic references was changed.

The specific characteristics of the iMillennials generation were initially described in table 1 with references to 21-23. The corrected bibliographic references are 28-29.

C.14. Conclusions very simple, underdeveloped. They do not contribute much to the development of the education system for iMillennials in the era of Industry 4.0. They do not respond to signals about the aims of work mentioned quite generally in Abstract and Introduction.

Developing them and enriching them with specific proposals for changes in the iMillennials generation system and education methods would significantly increase the value of work.

Some new comments were inserted in conclusions:

In this paper, the focus was on identifying the level at which some positive main characteristics of iMillennials could be quantified in order to improve them through special programs. Lack of concentration being a negative element was not quantified. For a group of students with technical background, two psychological inventories that describe emotional intelligence and motivational acquisitions were applied. Each inventory used was focused on certain features that describe motivational achievement (AMI) or emotional intelligence (EQ-I). As a result, a correlation between three parameters (assertiveness, testing reality and commitment) was determined. Based on these results the constellation map of soft skills occurred matching characteristic features of the iMillennials and necessary competences for an Industry 4.0 environment. Further research will focus on the various applications of this constellation map of soft skills. We intend to also consider the inclusion of the negative features of iMillennials. The approach from this perspective will constitute one of the further developments of our research, especially from the perspective of educational policies.

C 15. There is no legend for the description. Figure 6. It is not known whether graphics mean stronger meaning of some skills. Stars of different shapes can mislead the reader if they mean nothing. It is necessary to make up for this lack. It is also worth developing a comment to the map in Figure 6. A good comment with relevant recommendations on how to realistically use such ordered knowledge about the generation of iMillenials would increase the value and reception of work.

This supplementary comment was inserted:

“Starting from the vision of the main meta-factors, around which orbit the secondary factors, we tried to define the constellations of abilities and personal qualities - soft-skills. We designed three meta-factors (primary valences): Emotional Intelligence, Communication Strategies and Thinking outside the box. Around each of these primary valences gravitates a series of substructures described as secondary valences. For each new job involved by the Industry 4.0 paradigm, certain specific secondary factors could be selected so that to fit the job description. In order to calibrate and test that map of skills and personal qualities - soft-skills, we assessed the Industry 4.0 capabilities for a batch of students with an academic profile in the field of engineering sciences.”

C16. In Conclusions, there was a reference to Educational Policies - line 226, previously, unfortunately, little was said about it. It is worth to develop this topic in a new, expanded version of the Introduction to tie it in a coherent way with the Conclusions.

The introduction was expanded and linked to conclusions (see C3, C4).

C17. The literature review should be expanded and updated.

The literature review was expanded and updated (see C3).

C18. Please check for some editorial errors, missing references. Moreover, inconsistency in text format and reference style. Please present the references according to Education Sciences guidelines.

We made the suggested improvements accordingly.

Reviewer 3 Report

see attachment
